# Role of Fat-Free Mass Index on Amino Acid Loss during CRRT in Critically Ill Patients

**DOI:** 10.3390/medicina59020389

**Published:** 2023-02-17

**Authors:** Vaidas Vicka, Alvita Vickiene, Sigute Miskinyte, Ieva Bartuseviciene, Ingrida Lisauskiene, Mindaugas Serpytis, Donata Ringaitiene, Jurate Sipylaite

**Affiliations:** 1Clinic of Anaesthesiology and Intensive Care, Institute of Clinical Medicine, Faculty of Medicine, Vilnius University, LT-03101 Vilnius, Lithuania; 2Vilnius University Hospital Santaros Klinikos, LT-08661 Vilnius, Lithuania; 3Clinic of Gastroenterology, Nephro-Urology and Surgery, Institute of Clinical Medicine, Faculty of Medicine, Vilnius University, LT-03101 Vilnius, Lithuania; 4Faculty of Medicine, Vilnius University, LT-03101 Vilnius, Lithuania

**Keywords:** amino acid loss, CRRT, bioelectrical impedance analysis, FFMI

## Abstract

*Background and objectives:* Amino acid (AA) loss is a prevalent unwanted effect of continuous renal replacement therapy (CRRT) in critical care patients, determined both by the machine set-up and individual characteristics. The aim of this study was to evaluate the bioelectrical impedance analysis-derived fat-free mass index (FFMI) effect on amino acid loss. *Materials and methods:* This was a prospective, observational, single sample study of critical care patients upon initiation of CRRT. AA loss during a 24 h period was estimated. Conventional determinants of AA loss (type and dose of CRRT, concentration of AA) and FFMI were entered into the multivariate regression analysis to determine the individual predictive value. *Results*: Fifty-two patients were included in the study. The average age was 66.06 ± 13.60 years; most patients had a high mortality risk with APAHCE II values of 22.92 ± 8.15 and SOFA values of 12.11 ± 3.60. Mean AA loss in 24 h was 14.73 ± 9.83 g. There was a significant correlation between the lost AA and FFMI (R = 0.445, B = 0.445 CI95%: 0.541–1.793 *p* = 0.02). Multivariate regression analysis revealed the independent predictors of lost AA to be the systemic concentration of AA (B = 6.99 95% CI:4.96–9.04 *p* = 0.001), dose of CRRT (B = 0.48 95% CI:0.27–0.70 *p* < 0.001) and FFMI (B = 0.91 95% CI:0.42–1.41 *p* < 0.001). The type of CRRT was eliminated in the final model due to co-linearity with the dose of CRRT. *Conclusions:* A substantial amount of AA is lost during CRRT. The amount lost is increased by the conventional factors as well as by higher FFMI. Insights from our study highlight the FFMI as a novel research object during CRRT, both when prescribing the dosage and evaluating the nutritional support needed.

## 1. Introduction

Acute kidney injury (AKI) is reported in up to 60% of intensive care unit (ICU) patients [1]. It is known that AKI affects protein degradation and amino acid conversion due to impaired kidney metabolic functions. Therefore, patients with AKI have decreased serum amino acid concentrations [2]. Around 23% of AKI cases in the ICU are severe and life-threatening and may require continuous renal replacement therapy (CRRT) to remove metabolic waste products and to regulate fluid and electrolyte balance [1]. Inevitably, the mechanisms by which CRRT removes unwanted substances will also remove some essential solutes such as micronutrients and amino acids [3]. Various studies have reported that, during one day of CRRT, from 14 to 22 g of various amino acids are lost, which contributes to a negative nitrogen balance and increases the patient’s protein needs [2,3,4]. These losses, if not accounted for, are associated with worse clinical outcomes [5,6].

To improve clinical outcomes, CRRT needs to be patient specific to meet their dynamic needs. Historically, the evidence behind CRRT delivery has been derived from studies in which aspects of CRRT delivery have been investigated under the assumption that the optimal prescription of CRRT will be the same in all patients at all points during their need for renal replacement therapy [7]. Accordingly, while there are studies on baseline standards of CRRT delivery, uncertainty remains as to how to best adapt and personalize it to meet the requirements of specific clinical scenarios [8,9]. The most common variables of CRRT that can be adapted are the type, dose and length of the therapy, which are known to impact the loss of amino acids, but not account for it entirely. Therefore, personalized determinants of amino acid loss should be considered [3]. Consequently, the CRRT regimen could be enhanced, delivered and monitored to meet the individual solute and fluid requirements of patients.

Previous studies have demonstrated that critically ill patients can lose up to 17–30% of their muscle mass within the first 10 days of ICU admission, which is crucial as the major depot of amino acids in the body is in the muscles [10]. During critical illness, many patients enter a hypercatabolic state. This state causes an increased production of stress mediators including cytokines, hormones and immune mediators, which in turn activate cell catabolism: glycogenolysis, gluconeogenesis, lipolysis and, most importantly, proteolysis. Proteins are catabolized and amino acids are released from the muscles to the blood as a substrate for gluconeogenesis, which in turn lowers the blood amino acid concentration [2]. Therefore, if the patient is present with low muscle mass, it could indicate a higher risk of CRRT-induced amino acid loss, rendering muscle mass as an object of investigation. The European Society for Clinical Nutrition and Metabolism (ESPEN) recommends bioelectrical impedance analysis (BIA) and, in particular, BIA-derived fat-free mass (FFM) [11]. However, no studies have yet reported a link between BIA-derived FFMI and loss of amino acids during CRRT.

Therefore, the aim of this study was to evaluate the FFMI of the patients upon the initiation of CRRT and to determine the importance of it on amino acid loss.

## 2. Materials and Methods

### 2.1. Patient Sample

This was a prospective, observational, single sample study of critical care patients upon the initiation of CRRT, conducted at the tertiary reference university hospital during the period from March 2021 to June 2022. The inclusion and exclusion criteria are presented in Table 1. The patient sample was calculated according to the assumed variability of the amino acid amount lost during CRRT, suggesting a sample of at least 50 patients.

Ethical approval (No. 2021/2-1306-784) was granted from the Vilnius Regional Biomedical Research Ethics Committee to conduct the study. Written consent was obtained from the next of kin of all of participants in the study.

### 2.2. Measurement of Amino Acid Loss

Amino acid loss was calculated by using samples from the effluent fluid, obtained after the initiation of CRRT. The first samples were obtained during the first 24 h of the CRRT procedure, the sampling was carried out simultaneously from the incoming, outgoing and effluent lines of the extracorporeal circuit. During a period of 24 h, the parameters of the CRRT machine were not changed. There was no intervention in the set-up of the CRRT, it was conducted according to the standard practice of the hospital. The modalities used during the study were continuous veno-venous hemodialysis (CVVHD), continuous veno-venous hemodiafiltration (CVVHDF) and continuous veno-venous hemofiltration (CVVH). The initial dose for the first 24 h of CRRT was 30 mL/kg/h.

Calorimetric assay was used to determine amino acid concentrations. Before measuring the concentration of amino acids in human blood with the L-amino acid quantification kit, all samples were diluted 5 times by mixing 20 µL of blood with 80 µL of working buffer solution from the kit. Subsequently, the diluted samples were thermally inactivated by heating them at 90 °C for 15 min. After thermal inactivation, the samples were left to reach room temperature and centrifuged for 15 min at 15,000× *g* to remove the precipitate. The concentration of L-amino acids was measured according to the supplier’s technical bulletin (Sigma-Alrich product code MAK002-1KT). Briefly, 50 µL of sample was placed in the well of a 96-well plate and mixed with 50 µL of the master mix (composed of working buffer solution, enzyme mixture and a probe) and incubated at 37 °C for 30 min. After incubation, the absorbance at a 570 nm wavelength was measured, compared to the blank, and the concentration of L-amino acids was calculated according to the calibration curve. The calibration curve was also obtained by mixing 50 µL of standard amino acid solutions (0, 0.16, 0.32, 0.48, 0.64, 0.8 mM) with 50 µL of the master mix.

Individual net volume in liters of effluent fluid during the 24 h period was calculated. The individual moles of amino acids lost were calculated and by employing the average molecular weight of an amino acid of 110 Da, converted into grams lost per 24 h. The formula for grams lost per 24 h is as follows:$$Amino\;acids{\left(\frac{mol}{L}\right)}_{effluent}*Volume{\left(L\right)}_{effluent}$$
$$*Molecular\;weight\;{\left(110\;g/mol\right)}_{amino\;acids}$$

### 2.3. Measurements of Bioelectrical Impedance Analysis

BIA was performed to all the patients after enrolling to the study, i.e., upon the start of the CRRT. An InBody 72 S10 device (Biospace, Seoul, Republic of Korea) was used, following the instructions provided by the European Society for Clinical Nutrition and Metabolism [12]. Prior to analysis, the patients were positioned in a lying posture for 10 min. During the analysis, the patients were in a supine position with arms abducted 15 degrees from the trunk and legs spread apart at shoulder width. The analysis was performed using eight electrodes placed on both hands (on thumb and middle finger) and between the patient’s anklebones and heels. BIA provides resistance and reactance measured using different frequency currents in five different segments of the body: both arms, both legs and the trunk. These impedance values are when plotted against the reference values of the healthy population and thus body composition is estimated. The measurements obtained were fat mass, fat-free mass, intracellular water, extracellular water, total body water and phase angle. The fat-free mass was further indexed by the height of the patient (FFMI).

### 2.4. Statistical Analysis

#### 2.4.1. Descriptive Analysis

Statistical analysis was carried out by the SPSS statistical software package version 23.0 (IBM/SPSS, Inc., Chicago, IL, USA). Baseline characteristics were defined using descriptive statistics. Categorical variables were stated as an absolute number (n) and a relative frequency (%), and continuous variables were represented as a median (interquartile range) or as a mean (± SD), depending on the normality of the distribution. The normality of the distribution was tested by the single sample Kolmogorov–Smirnov test.

#### 2.4.2. Association between the Fat-Free Mass and Lost Amino Acids Lost

To determine the association between the FFMI and AA lost, linear regression was performed, providing regression coefficients.

#### 2.4.3. Regression Analysis

Conventional factors and BIA-derived FFMI were entered into the multivariate regression model. Variables significantly associated with the lost AA in the univariate model were further entered into the multivariate regression model. The final multivariate model was tested for accuracy and collinearity.

## 3. Results

### 3.1. Baseline Characteristics

Fifty-two patients were enrolled in the study. These patients were a high mortality risk with high APACHE II and SOFA scores, medium age of the patients was 66 years. The sample was a mix cohort between the surgical (44%) and therapeutic (56%) patients.. The description of the sample is presented in Table 2.

### 3.2. Amino Acid Loss

Two of the patients included in the study were eliminated from the analysis due to hemolysis in the samples and false results above the calibration curve. The mean amino acid concentration in the circulation upon the initiation of CRRT was 2.20 ± 0.97 mmol/L. The median concentration in the effluent fluid was 1.86 [1.26–2.87] mmol/L. The mean amount of AA lost per 24 h was 14.73 ± 9.83 g.

### 3.3. Fat-Free Mass and Association with Amino Acid Loss

Bioelectrical impedance analysis was performed for 50 of the 52 patients because two of the patients did not have all four limbs. The median FFMI mass index of the cohort was 22.82 ± 3.90 kg/m^2^, which is above the cut-off value for both men and women for nutritional risk. The results on hydration state of the patients show a moderate fluid overload, with increased total body water (TBW), extracellular body water (EBW) and higher oedema index (ECW/TBW) of 0.43. The phase angle, calculated from the raw data of resistance and reactance from the measurements of the 50 kHz electric current, also tends to be lower with mean value of 3.9, which is below the age, gender and body mass index adjusted normal values of the cohort. This may be indicative either of fluid overload, or malnutrition of the patients. The results of BIA are presented in Table 3.

Correlation analysis was carried out to determine the relationship between the FFMI and AA lost. A moderate correlation was found between the two variables (R = 0.445, B = 0.445 CI95%: 0.541–1.793 *p* = 0.02) (Figure 1).

### 3.4. Regression Analysis of Lost AA Determinants

Regression analysis was started with the univariate regression. In univariate regression co-morbidities, severity indices, amino acid concentration, bioelectrical impedance measurements and CRRT set-up parameters were added. The only variables associated with lost AA amount were fat-free mass index, systemic amino acid concentration and the parameters of the CRRT set-up (modality and the dose). These variables were added to the multivariate regression. However, due to collinearity between the dose and type of the CRRT, the type of CRRT was removed from the final regression model. The final regression model was accurate (R = 0.806, R2 = 0.625, *p* = 0.01), with no signs of collinearity (Table 4).

The regression analysis revealed the dose of the dialysis as one of the factors increasing the lost AA, per 10 mL/kg/h of increase in dose leading to increase of 4.84 g in lost AA. The type of the dialysis was also important, with mean lost AA of 7.15 ± 2.54 g for CVVH, 11.92 ± 7.21 g for CVVHD and 20.78 ± 11.40 g for CVVHDF (*p* = 0.02). However, due to collinearity between dialysis dose and modality type, the modality type had to be removed from the final regression model. These results are reported from the univariate regression.

The systemic concentration of AA had a strong correlation with the AA lost, per 1 mmol/L increase in systemic concentration leading to 7.00 g of lost AA.

The FFMI also prevailed as an independent predictor of lost AA, as a 5 kg/m^2^ increase in FFMI led to the loss of 4.57 g of AA.

## 4. Discussion

Our study of critically ill AKI patients on CRRT reported that the mean AA loss during the first 24 h of a CRRT session was almost 15 g. The amount of AA lost was determined by both the dose and the type of CRRT, by the systemic concentration of AA and by the fat-free mass index of the patient.

The amount of lost AA in our study is similar to the findings of other studies that reported AA loss of 5–15 g/day. These results yet again signify that AA loss during CRRT is considerable and may contribute to a negative protein balance that may worsen critically ill patient outcomes [2]. In our study, both the dose and the type of the CRRT were significant predictors of AA loss, which is also concordant with the current literature [5]. Regarding the dialysis dose, recent studies in ICU patients have pointed out the positive correlation between CRRT dose and magnitude of AA losses and determined that large ultrafiltration rates result in significant negative nitrogen balance [13]. As for the modality of CRRT, we could not provide a definitive answer in our study, since the general practice in our centre is to use CVVH when a small dosage of CRRT is needed and to use CVVHDF when a large dosage is needed. Therefore, we could not avoid the collinearity between the type and dose. However, it has been reported that, during CVVH, clearance is mainly achieved by convection. Due to the non-selectivity of the filtering process, substantial parts of blood are also lost, including amino acids, driving higher protein losses during CVVH than during CVVHD/F (6.0 mg/dL versus 2.7 mg/dL) [14]. On the other hand, another study that found up to 12% loss of dietary amino acid intake in patients on CRRT found no significant difference regarding the modality [15]. These discrepancies in the literature can either be explained by the dosage, or by taking into the account the individual parameters of the patients.

One of the individual parameters described as a determinant of AA loss in our study was the systemic AA concentration. Critically ill patients with AKI are often hypercatabolic, with increased protein catabolism and mobilization of AA to the systemic circulation as a substrate for gluconeogenesis. However, even though the net amount of AA production is higher, the systemic concentration tends to be lower [16]. In our study, the overall systemic concentration of AA was lower than in healthy people and comparable to the general ICU population, with a strong correlation between a higher AA concentration and higher loss of AA during CRRT [17]. This is a likely finding, since concentration gradient is one of the determinants of molecule diffusion across a semi-permeable membrane. This relationship is supported by previous studies demonstrating that the administration of enteral feeding may increase protein losses by increasing the serum levels of relatively modestly sized proteins that may be of a filterable molecular weight. Furthermore, when these nutritional supplements are infused during dialysis, losses are increased, but a net positive balance can be achieved if the infusion rate is high enough [18]. However, these studies do not account for the intrinsic massive storage of AA in patients’ bodies, i.e., muscle mass.

The most reliable, well-validated non-invasive and relatively inexpensive assessment of body composition is bioelectric impedance analysis, recommended by the ESPEN, ASPEN, as well as the Global Leadership Initiative on Malnutrition. One of the BIA parameters, fat-free mass index, has potential use in the estimation of metabolic rate, protein requirements and pharmacokinetics. Previous studies show that higher lean mass index is associated with higher concentrations of branched-chain amino acids in both sexes [19]. Our study suggests that higher fat-free mass index is associated with higher AA loss during CRRT. This can be partially explained by the fact that fat-free mass serves as a depot of AA in the organism. However, when performing the multivariate regression analysis, we found that FFMI remained an independent predictor of AA loss, and even then, entered in the same model as the systemic AA concentration. This suggests that the high content of muscle in the tissues may be more readily mobilized during a hypercatabolic critical state and consequently more readily eliminated via extracorporeal clearance. These results are inconclusive when transferred to clinical practice, as intuitively it seems that a higher loss of AA during CRRT would worsen the clinical outcome. However, it may be a protective mechanism to endure a prolonged critical state by the employment of larger reserves in proteins and AA and not allowing the AA concentration to fall too low. These insights from our study highlight the FFMI as a novel research object during CRRT, both when prescribing the dosage and when evaluating the nutritional support needed.

This was a fundamental study, designed to shed some insights of the metabolism of critical care patients. Therefore, clinical outcomes have not been registered and accounted for. However, with the results of the study, some possible clinical implications have become evident and are worth discussing. Firstly, the absolute amount of amino acids lost reported in our study is substantial and dictates clinical intervention with supplementation. The reported 15 g of lost amino acids per day is not big; however, in some cases, this amount was up to 40 g. In these cases, especially if low FFMI is present, the ESPEN recommended amount of protein per day may not be enough. Furthermore, when looking for predictors of worse clinical outcome the cumulative loss would be a more appropriate marker. Secondly, this study has some clinical insights for the technology and application of CRRT. The most important finding is conceptual: we must evaluate the specific individual needs of the patients, and not use “one size fits all” approach. In this study, FFMI and amino acid concentration were proven to be instruments of personalized medicine. Lastly, this study emphasizes the importance of body constitution of the patients in the beginning of the critical illness. The results of our study implicate the importance of fat-free mass, i.e., mostly the muscle tissue. There is no clinical intervention, which would promote the muscle growth in the beginning of the critical state, apart from the treatment of the underlying disease. Therefore, from a boarded perspective, it adds to an increasing pool of evidence on worse clinical outcomes in a frail and sarcopenic aging society.

This study has some limitations. It was a small sample study with heterogeneous pathologies; the complexity of critical illnesses may have led to residual confounding factors. Firstly, our study selection criteria were not specific, so patients in the study group may be uneven in terms of metabolism and hypercatabolic drive. In this study, there are no SIRS markers reported (TNF alfa, IL-1, IL-6, etc.). The addition of the biomarkers may give more insight into the mobilization of amino acids from the muscle reserve and a consequential loss during CRRT. However, the study was planned to establish the well-known CRRT machine set-up effect and to identify potential individual variables associated with amino acid loss. Only upon interpreting the results, the importance of muscle reserve and mobilization of amino acids became evident. Secondly, AA sampling was performed only once a day, which may not be accurate for a dynamic ICU patient. Also, we did not include and compare results based on the diet regimen, glycemia status, insulin intake or adsorption of AA on the filter membrane, which may have influenced both systemic and lost AA concentrations. Additional data processing and complex analysis are required for further conclusions.

Furthermore, the general limitations of the bioelectrical impedance analysis apply. The main shortcoming of BIA is a disturbed hydration state and shortage of large reference studies in critical care. Because of the excess fluid, the measured impedance (reactance and resistance) is lower, and therefore the FFMI estimation is defective, mainly because the regression equations are based on a healthy population screening. While not a direct justification of these assumptions, the oedema index of 0.43 in the study indicates that fluid overload was not extensive. However, these limitations do not detract the importance of the bioelectrical impedance method in critical care. In our study, we did not evaluate the patients for malnutrition, or risk of malnutrition. Thus, exact numbers of FFMI, which are necessary for malnutrition diagnostics, were not used. FFMI in relative numbers was used to establish the connection to lost amino acid amount.

## 5. Conclusions

A substantial amount of AA is lost during CRRT and may advocate the need for clinical intervention. A conceptual finding of the study is that the amount of amino acids lost during CRRT is increased not only by the CRRT set-up (i.e., modality and the dose) but also by individual and specific parameters of the patients: by a higher systemic concentration of amino acids and by a higher FFMI. Insights from our study highlight the FFMI as a novel research object during CRRT, both when prescribing the dosage and when evaluating the nutritional support needed. Furthermore, more research should be carried out on FFMI as a reserve of AA during critical illness, focusing on AA mobilization from the tissues.

## Figures and Tables

**Figure 1 medicina-59-00389-f001:**
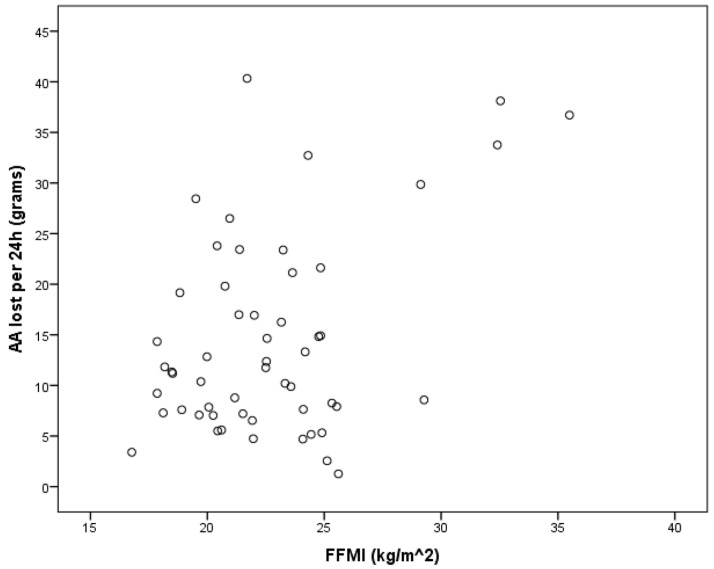
Relationship between the FFMI and AA lost. The x axis denotes the fat-free mass (FFMI) and the y axis denotes the amino acid (AA) lost per 24 h. The dots denote individual values.

**Table 1 medicina-59-00389-t001:** Selection criteria.

Inclusion Criteria	Exclusion Criteria
Start of CRRT	Not eligible for BIA
Mechanical ventilation	Expected lethal outcome in <48 h
	Age <18 y

CRRT—continuous renal replacement therapy, BIA—bioelectrical impedance analysis.

**Table 2 medicina-59-00389-t002:** Description of the patients.

	Mean ± SD, Median IIQR], n (%)
Demographics	
Age	66.06 ± 13.60
Gender:	
Male	22 (42.3)
Female	30 (57.7)
Co-morbidities	
Arterial hypertension	41 (78.8)
Heart failure	29 (55.8)
Diabetes	12 (23.1)
Chronic kidney disease	3 (5.8)
Chronic obstructive pulmonary disease	1 (1.9)
Immunosuppression	3 (5.8)
Severity indices	
APACHE II	22.92 ± 8.15
SOFA	12.11 ± 3.60
On vasopressors	50 (96.2)
Mean dose of noradrenaline (µg/kg/min)	0.25 [0.1–0.41]
CRRT parameters	
Initial dose (mL/kg/h)	30.12 ± 8.77
Modality:	
CVVHD	30 (57.7)
CVVH	4 (7.7)
CVVHDF	18 (34.6)
Disease course and outcomes	
Therapeutic profile	29 (55.8)
Surgical profile	23 (44.2)
Days before ICU	6 [2–12.5]
Hospital stay	24 [10–42.5]
ICU stay	12 [5.8–20.5]

SD—standard deviation, IQR—interquartile range, SAPS—Simplified Acute Physiology Score, SOFA—Sequential Organ Failure Assessment, APACHE II—Acute Physiology and Chronic Health Evaluation, ICU—intensive care unit, CRRT—continuous renal replacement therapy, CVVH—continuous veno-venous hemofiltration, CVVHD—continuous veno-venous hemodialysis, CVVHDF—continuous veno-venous hemodiafiltration.

**Table 3 medicina-59-00389-t003:** Bioelectrical impedance analysis.

	Mean ± SD
Weight (kg)	87.70 ± 21.23
Height (cm)	173.0 ± 8.12
FFM	68.30 ± 14.05
FFMI	22.82 ± 3.90
ICW	29.79 ± 6.00
ECW	21.46 ± 4.76
TBW	50.46 ± 11.67
Phase angle	3.88 ± 1.15

FFM—fat-free mass, FFMI—fat-free mass index, ICW—intracellular water, ECW—extracellular water, TBW—total body water.

**Table 4 medicina-59-00389-t004:** Regression analysis of the lost AA determinants.

	Multivariate Regression			
Variable	B	Exp (B)	95% CI for B	*p* Value
Dialysis dose	0.484	0.439	0.269–0.700	<0.001
Systemic concentration of AA	6.999	0.672	4.956–9.042	0.001
FFMI	0.913	0.362	0.417–1.408	<0.001

B—regression coefficient B, CI—confidence interval, AA—amino acids, FFMI—fat-free mass index.

## Data Availability

The dataset used during the current study is available from the corresponding author on reasonable request according to the data protection policies.

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
