# Peer review of "Role of Fat-Free Mass Index on Amino Acid Loss during CRRT in Critically Ill Patients"

_medicina, 2023, doi:10.3390/medicina59020389_

Round 1

Reviewer 1 Report

Nutrition in critical patients is surely a fundamental issue so I'm pleased to read your paper. Nevertheless, I would you to adress some potential clinical solutions according to your experience to reach an higher clinical relevance.

Reviewer 2 Report

Amino acid loss in patients with critical illness is associated  with many factors, as the author indicated in the introduction section. The nutritional risk of critically ill patients needs to be comprehensively evaluated from the aspects of chief complaint and past medical history, physical examination/clinical signs, anthropometric data, laboratory indicators, functional evaluation and evaluation scale. Especially when the hydration state of the body changes under the abnormal condition, bioelectrical impedance analysis is not very accurate in estimating the fat-free mass of the body. Therefore, bioelectrical impedance method alone is used to evaluate the fat-free mass index (FFMI) in patients with critically ill caused by multiple factors. I suggest that it should add other factors including traditional serum protein markers, interleukin-1, tumor necrosis factor and interleukin-6 in the multivariate regression analysis. In addition, The mechanism of the relationship between FFMI and amino acid loss should be studied or discussed.
